# Monitoring Wheat Growth Using a Portable Three-Band Instrument for Crop Growth Monitoring and Diagnosis

**DOI:** 10.3390/s20102894

**Published:** 2020-05-20

**Authors:** Huaimin Li, Weipan Lin, Fangrong Pang, Xiaoping Jiang, Weixing Cao, Yan Zhu, Jun Ni

**Affiliations:** 1College of Agriculture, Nanjing Agricultural University, Nanjing 210095, China; 2019101174@njau.edu.cn (H.L.); 2018814037@njau.edu.cn (W.L.); pangfangrong@njau.edu.cn (F.P.); xpjiang@njau.edu.cn (X.J.); caow@njau.edu.cn (W.C.); yanzhu@njau.edu.cn (Y.Z.); 2National Information Agricultural Engineering Technology Center, Nanjing 210095, China; 3Engineering Research Center of Smart Agriculture, Ministry of Education, Nanjing 210095, China; 4Jiangsu Collaborative Innovation Center for the Technology and Application of Internet of Things, Nanjing 210095, China

**Keywords:** multispectral sensor, crop growth status, vegetation index, growth period, spectral monitoring model, precision agriculture, agricultural remote sensing

## Abstract

An instrument developed to monitor and diagnose crop growth can quickly and non-destructively obtain crop growth information, which is helpful for crop field production and management. Focusing on the problems with existing two-band instruments used for crop growth monitoring and diagnosis, such as insufficient information available on crop growth and low accuracy of some growth indices retrieval, our research team developed a portable three-band instrument for crop-growth monitoring and diagnosis (CGMD) that obtains a larger amount of information. Based on CGMD, this paper carried out studies on monitoring wheat growth indices. According to the acquired three-band reflectance spectra, the combined indices were constructed by combining different bands, two-band vegetation indices (NDVI, RVI, and DVI), and three-band vegetation indices (TVI-1 and TVI-2). The fitting results of the vegetation indices obtained by CGMD and the commercial instrument FieldSpec HandHeld2 was high and the new instrument could be used for monitoring the canopy vegetation indices. By fitting each vegetation index to the growth index, the results showed that the optimal vegetation indices corresponding to leaf area index (LAI), leaf dry weight (LDW), leaf nitrogen content (LNC), and leaf nitrogen accumulation (LNA) were TVI-2, TVI-1, NDVI (R_730_, R_815_), and NDVI (R_730_, R_815_), respectively. R^2^ values corresponding to LAI, LDW, LNC and LNA were 0.64, 0.84, 0.60, and 0.82, respectively, and their relative root mean square error (RRMSE) values were 0.29, 0.26, 0.17, and 0.30, respectively. The addition of the red spectral band to CGMD effectively improved the monitoring results of wheat LAI and LDW. Focusing the problem of vegetation index saturation, this paper proposed a method to construct the wheat-growth-index spectral monitoring models that were defined according to the growth periods. It improved the prediction accuracy of LAI, LDW, and LNA, with R^2^ values of 0.79, 0.85, and 0.85, respectively, and the RRMSE values of these growth indices were 0.22, 0.23, and 0.28, respectively. The method proposed here could be used for the guidance of wheat field cultivation.

## 1. Introduction

Wheat (*Triticum aestivum* L.), which is the oldest cultivated crop, is presently an important staple crop in China; thus, improving the level of wheat production is critical to ensuring national food security and economic income to farmers [1]. With the development of agricultural cultivation techniques and the improvement of the level of agricultural mechanization, wheat production and management has gradually shifted to being more precise and intelligent; thus, effectively obtaining wheat growth information is an important prerequisite for precision agricultural operations [2]. Although traditional laboratory analysis methods can accurately obtain the physiological and biochemical parameters of wheat, these methods are laborious, time-consuming, and cannot be applied in agricultural production. In recent years, the development of spectral monitoring techniques has made it possible to obtain wheat growth information in real-time and in a rapid, non-destructive manner [3].

Based on spectral monitoring techniques, Chinese and international research institutes have developed a variety of multi-spectral and hyper-spectral crop growth monitoring apparatuses based on different platforms and have carried out a series of experimental studies using those apparatuses. Portable instruments are a popular choice in the field of crop growth monitoring for their convenience and portability. Single-leaf scale monitoring equipment is currently the most mature of these apparatuses. Uddling et al. [4] monitored the chlorophyll content of wheat based on the chlorophyll-content monitor SPAD-502 (Konica-Minolta, Tokyo, Japan), and the obtained readings had a strong non-linear correlation with the chlorophyll content of wheat leaves, which could achieve the diagnosis of wheat chlorophyll in different cultivars and growth stages. Cerovic et al. [5] performed experimental studies on crops, such as wheat and corn, using the multi-functional leaf measuring instrument Dualex 4 (FORCE-A, Orsay, France). The research results showed that the instrument could accurately obtain the chlorophyll index, flavonoid index, and nitrogen balance index of crop leaves, achieve the fast and non-destructive monitoring of crop growth, and help crop field nitrogen management. These two monitors have high accuracy, but can only monitor indices, such as the chlorophyll content of crop leaves, by point sources. They cannot represent the growth index of crop populations, and their application is limited.

Canopy-scale crop growth monitoring equipment can obtain vegetation indices to build crop growth monitoring models, with the advantages of having a wide monitoring range and being able to monitor multiple indices; it is currently the main type of monitoring equipment on the market. Zhang et al. [6] used a GreenSeeker portable spectrometer (Trimble Navigation Limited, Sunnyvale, CA, USA) to monitor the growth indices of winter wheat, constructed the new vegetation indices rNDVI and rRVI based on the NDVI and RVI obtained by the instrument, and realized effective monitoring of indices such as leaf area index (LAI) and biomass. However, the retrieval effect of leaf nitrogen content (LNC) at the jointing stage of wheat was poor. Osborne [7] used a GreenSeeker portable spectrometer to estimate in-season growth and N status in spring wheat and found that NDVI obtained by this sensor showed a significant relationship with plant biomass, N concentration, and N uptake. The result suggested that GreenSeeker could also be used in the northern Great Plains for estimating in-season N need for spring wheat. Cao et al. [8] used Crop Circle ACS 470 (Holland Scientific Inc., Lincoln, NE, USA) to estimate the rice yield potential in a growing season and found that the device had relatively high monitoring accuracy at the jointing, booting, and heading stages (R^2^ values were greater than 0.7), and the device could effectively improve rice field nitrogen management. Cao et al. [9] used ACS 470 to monitor plant N uptake (PNU) and grain yield of winter wheat by constructing 43 vegetation indexes. The results indicated that the decision coefficients of PNU and grain yield models based on ACS 470 were 0.78 and 0.62, respectively, and the prediction accuracy was significantly better than that of GreenSeeker. Feng et al. [10] used an ASD FieldSpec Pro FR 2500(Analytical Spectral Devices, Boulder, CO, USA) to obtain wheat canopy hyperspectral data and compared the monitoring effects of different sensitive bands and spectral indices on wheat nitrogen indices. The results showed that the optimal vegetation indices corresponding to LNC and leaf nitrogen accumulation (LNA) were REIPle and FD452, respectively, and the prediction accuracies reached 0.75 and 0.87, respectively. While the above spectrometers can monitor different crop growth indices, they still have some problems in agricultural applications. Some of the commercial instruments mentioned cost more than CNY 100,000 (about USD 14,000), which is difficult for most farmers to accept. In addition, hyperspectral instruments such as ASD simultaneously acquire reflectance of hundreds of bands, and the data redundancy is large. These instruments require professional knowledge of data processing to extract effective information, which limits their promotion. 

In response to the problems of these instruments, Nanjing Agricultural University developed a portable instrument of crop-growth monitoring and diagnosis (CGMD) CGMD-302, which can effectively obtain a variety of growth indices, such as crop LAI, biomass, and nitrogen status, and has the advantages of being easy to carry, cost-effective, and easy to operate [11,12]. However, the CGMD-302 contains only two bands, red edge and near-infrared, and it only obtains one vegetation index. Due to the lack of visible bands of light, there may be insufficient spectral information and low accuracy of the retrieved growth index. Chu et al. [13] pointed out that the spectral index of combined red edge and near-infrared bands is more sensitive to the LNA, but it is not suitable for monitoring LNC. Previous studies have shown that there are differences in the sensitive bands corresponding to different growth indices. The bands in the vicinity of 550 nm and 650–760 nm can effectively reflect the nitrogen status of plants. The spectral reflectance values near 450, 550, 670, and 870 nm show a high correlation with LAI, and the bands near 560, 670, 810, and 930 nm are more sensitive to biomass indices [14,15,16,17,18]. The visible light region is highly correlated with several growth indices and is the core band used in crop spectral monitoring. According to the photochemical characteristics of crops, chlorophyll a and b show the maximum absorption in the red band. Previous studies have found a strong relationship with crop characteristics is located in specific narrow bands in the portion of red [19,20]. Bunnik [21] found that the spectral characteristics of the red band of the crop canopy are different from those of the near-infrared band, which can effectively supplement the information obtained by the instrument. Zhu et al. [22] also found that RVI (R_660_, R_810_) and RVI (R_660_, R_870_) were most highly correlated with LNA in both wheat and rice. In addition, existing studies have shown that the monitoring results of a three-band vegetation index on some growth indices is better than that of two-band vegetation indices, and a proper increase in the number of spectrometer bands is conducive to improving the monitoring accuracy [23,24,25,26]. Therefore, multi-band crop growth monitoring equipment equipped with red, red edge, and near-infrared bands is more consistent with the needs of crop growth monitoring.

To the CGMD, the red spectral band was added to the two bands of the CGMD-302, and as a result, the new instrument can simultaneously obtain the reflectance of red, red edge, and near-infrared spectral bands. According to the modeling needs of the growth indices of different crops, the appropriate vegetation index was then able to be selected, which allows for more flexibility and applicability in practical applications. Based on CGMD, the canopy reflectance values of wheat in red, red edge, and near-infrared spectral bands were obtained in this paper. A series of two-band and three-band vegetation indices was constructed, and the fitting relationship between vegetation indices and growth parameters was studied. A method of constructing the wheat-growth-index spectral monitoring models that were defined according to the growth periods is proposed in this study.

## 2. Materials and Methods

### 2.1. Design of Field Experiments

Wheat field experiments were conducted from November 2018 to April 2019 at the demonstration base of the National Engineering and Technology Center for Information Agriculture in Rugao City, Jiangsu Province, China (120°46’ E, 32°16’ N). The experimental cultivar used was Shengxuan No. 6. The three nitrogen fertilizer levels used in the experiment were the following: N0 (0 kg/hm^2^), N1 (180 kg/hm^2^), and N2 (360 kg/hm^2^). The ratio of base fertilizer to fertilizer at the jointing stage was 5:5. Two types of base fertilizer were applied simultaneously (135 kg/hm^2^ of P_2_O_5_ and 220 kg/hm^2^ of K_2_O). Three planting densities were used, with line spacings of D1 (20 cm), D2 (35 cm), and D3 (50 cm). A split-plot design was used in this experiment, with the density level as the main plot and the nitrogen fertilizer level as the secondary plot. There were nine treatments and three replicates, for a total of 27 plots. The length and width of each plot was 6 m and 5 m, respectively, for a total plot area of 30 m^2^. Spectral data and growth indices were obtained simultaneously as four of the growth stages of wheat: jointing, booting, heading, and flowering. Pest and other management procedures followed local standard production practices in wheat.

### 2.2. Equipment

#### 2.2.1. Portable Three-Band CGMD Instrument

The CGMD instrument consisted of multi-spectral crop growth sensor, processor system, sensor holder, level, shielded cable and other components, as shown in Figure 1. Compared with other instruments, the monitor was cost-effective, easy to carry, and suitable for field operation. The parameters of some common instruments are shown in Table 1. The sensor size of CGMD was smaller than other instruments, which made it easier to carry in the field. Although CGMD weighs about 1.6 kg, the support bar allows the user to put the instrument on the ground during test instead of holding it at all times. The support bar could adjust the height of the sensor to meet the monitoring requirements at different growth periods, and the level equipped on the support bar could calibrate the angle of the sensor lens in time. The unique structure of the CGMD made its operation simple and labor-saving.

The multi-spectral crop sensor was structurally divided into an upward light sensor and a downward light sensor. The upward light sensor could receive sunlight radiation information at 660, 730, and 815 nm bands; the downward light sensor is composed of three detector lenses for detection at characteristic wavelengths of 660, 730 and 815 nm, respectively. The design of the upward light sensor and the downward light sensor could correct the spectral reflectance in real time, avoiding the influence of sunlight on the monitoring accuracy. Each detector lens consists of a spectral filter and photoelectric detectors. To obtain the reflectance values of the crop canopy at 660, 730 and 815 nm bands, the radiation information is processed after being converted to electrical signals through the photoelectric detector. Multispectral sensors are shown in Figure 2. The simple and unique structural design of CGMD led it to exhibit a better cost performance than other commercial instruments, so this instrument has high application potential.

#### 2.2.2. ASD FieldSpec HandHeld2 Spectroradiometer

ASD FieldSpec HandHeld2 spectroradiometer (Analytical Spctral Devices Inc., Boulder, CO, USA) was used to evaluate the data acquisition performance of the portable three-band CGMD instrument. This device has a 512-element photodiode array (PDA) detector, which uses sunlight as a light source, with a wavelength range of 325–1075 nm, a scanning time of 17 ms, and a spectral sampling interval of approximately 1.5 nm. The measuring site of the instrument was 1 m above the canopy, the field angle of the spectrometer was 25°, and the measurement range was a circular plane. The measured data were calibrated by a white Spectralon reference panel to obtain the reflectance data. The monitor uses sunlight as the light source, and the measurement should be conducted at noon under a clear sky or scattered clouds. This spectroradiometer has the advantages of having a high resolution, narrow wave band, and good continuity, and it is currently recognized as a good CGMD instrument on the market. This instrument is shown in Figure 3.

#### 2.2.3. LAI-2200C Plant Canopy Analyzer

The LAI-2200C plant canopy analyzer (Li-Cor, Lincoln, NE, USA) was used to acquire observed LAI. The device uses a “fisheye” optical sensor (148° vertical field-of-view and 360° horizontal field-of-view) to measure transmitted light at five angles above and below the vegetation canopy. The canopy structure parameters, such as LAI, mean leaf inclination angle, void ratio, and aggregation index were calculated. The LAI-2200C is based on the mature LAI-2000 technology platform and has a built-in GPS module. It can integrate GPS information and perform scattered light calibration, allowing the LAI-2200C to be suitable for the canopy measurement under any weather conditions. This instrument is shown in Figure 4.

### 2.3. Measurement Method

#### 2.3.1. Measurement of Spectral Data

Spectral measurement requires the measurement environment to be in good light conditions and the measured object to remain stationary. Therefore, the measurements in this paper were conducted at noon with no wind. The measurement times at the following stages, i.e., early jointing, late jointing, booting, heading, and flowering, and the canopy reflectance spectra were obtained at three bands (660, 730, and 815 nm). In the experiment, the ASD FieldSpec HandHeld2 spectroradiometer was used to simultaneously monitor the crop canopy reflectance spectra. Three sites were selected for monitoring in each plot, and the average of the monitoring results in each plot was calculated. More detailed data are available in the Appendix A.

#### 2.3.2. Determination of Growth Indices

While the spectral data were acquired, the growth indices of wheat were obtained by destructive sampling. In each plot, 50 cm representative wheat plants were selected, and samples were divided indoors according to plant organs. The LAI-2200C was used to obtain the LAI at the time of sampling. Three sites were selected to be monitored in each plot, and the average of the monitoring results for each plot was calculated. The samples were fixed at 105 °C for 30 min, then the plants were dried to constant weight at 80 °C, and the sample was weighed to obtain leaf dry weight (LDW). Each sample was pulverized, and its LNC was determined by the Kjeldahl method [27]. LNA was calculated using the following equation.
(1)LNA=LNC×LDW

### 2.4. Data Analysis Methods

#### 2.4.1. Vegetation Index Calculation

The two-band vegetation indices selected in this study included the Normalized Vegetation Index (NDVI), Ratio Vegetation Index (RVI), and Difference Vegetation Index (DVI) [28,29,30]. Additionally, this study independently constructed two three-band vegetation indices: TVI-1 and TVI-2.

The construction of the three-band vegetation indices was based on the common construction form of two-band vegetation indices. An additional band was added to the original vegetation index to make it retain the physical characteristics and increase the amount of information. Among them, the construction of TVI-1 used the construction principle and a form of NDVI as a reference, replacing the red band of light in NDVI with (Rred + Rred-edge) and multiplying the near-infrared band by 2 to make it numerically equivalent to (Rred + Rred-edge). In developing TVI-2, the construction principle was used as well as a form of DVI as a reference. Additionally, (Rred + Rred-edge) replaced the red band of light in DVI, and the near-infrared band was multiplied by 2 to make it numerically equivalent to (Rred + Rred-edge). The equation of the above vegetation indices are as follows.
(2)$$NDVI=(R_{\rho }{}_{1}-R_{\rho }{}_{2})/(R_{\rho }{}_{1}+R_{\rho }{}_{2})$$
(3)$$RVI=R_{\rho 1}/R_{\rho 2}$$
(4)$$DVI=R_{\rho 1}-R_{\rho 2}$$
(5)$$TVI-1=\;(2\times R_{\rho 1}-R_{\rho 2}-R_{\rho 3})/(2\times R_{\rho 1}+R_{\rho 2}+R_{\rho 3})$$
(6)$$TVI-2=2\times R_{\rho 1}-R_{\rho 2}-R_{\rho 3}$$

#### 2.4.2. Data Analysis

Regression analysis was used to construct the model for LAI, LDW, LNC, and LNA estimation using the Excel 2016 software (Microsoft Inc., Redmond, WA, USA). Seventy percent of the datasets was used to construct monitoring models, and the remaining 30 percent was used to verify the prediction accuracy of the models. The data of the training set and the verification set came from different plots. The vegetation indices constructed from the obtained crop canopy spectral information were fitted to agronomic parameters based on least square method, and crop growth monitoring models were constructed by exponential regression analysis. The coefficient of determination (R^2^) and relative root mean square error (RRMSE) were used to comprehensively evaluate the performance of the model. The range of R^2^ is between 0 and 1; the larger the value, the higher the prediction accuracy of the model; the smaller the RRMSE value, the smaller the data difference and the higher the prediction accuracy of the model. The R^2^ values were evaluated by Excel 2016 software, and the RRMSE values were evaluated by MATLAB R2017b software (The Math Works Inc., Natick, MA, USA). The equations used for data analysis are as follows:(7)$$R^{2}=\frac{SSR}{SST}$$
(8)$$RRMSE=\sqrt{\frac{\sum_{i=1}^{n}{\left(P_{i}-O_{i}\right)}^{2}}{n}}\times \frac{100}{O_{i}}$$
where SSR, SST, P_i_, O_i_, and *n* are the sum of regression squares, sum of residual squares, predicted value, observed value, and number of samples, respectively. 

## 3. Results

### 3.1. Evaluation of Data Acquisition Performance of the Portable Three-Band CGMD Instrument

To evaluate the performance of the CGMD instrument in obtaining vegetation indices, regression analysis was performed on the corresponding values of NDVI, RVI, DVI, TVI obtained by the CGMD instrument in each growth period. To facilitate the comparison and analysis of the fitting accuracy of the combined vegetation indices of different wavebands, the vegetation indices obtained by FieldSpec HandHeld2 and CGMD were normalized. As shown in Figure 5, the fitting results of the vegetation indices obtained by CGMD and the corresponding values obtained by the commercial instrument FieldSpec HandHeld2 have a linear relationship, but some differences can be seen in the fitting accuracy of different vegetation.

The correlation of fitting of each vegetation index with different band combinations was compared. The fitting results of NDVI (R_730_, R_815_) were better than those of NDVI (R_660_, R_815_) and NDVI (R_660_, R_730_). The R^2^ values were 0.83, 0.67, and 0.56, respectively, and the RRMSE values were 0.16, 0.10, and 0.13, respectively; the combination of 660 and 730 nm and 660 and 815 nm showed severe dispersion at low values. There were similar patterns for different band combinations for RVI and DVI. For the 660 nm/730 nm, 660 nm/815 nm, and 730 nm/815 nm combinations, the R^2^ values of RVI were 0.53, 0.66, and 0.82, respectively, and the RRMSE values were 0.33, 0.35, and 0.07, respectively; the R^2^ values of DVI were 0.21, 0.60, and 0.79, respectively, and the RRMSE values were 0.21, 0.60, and 0.79, respectively. The accuracies of vegetation indices of different bands obtained by CGMD were in a descending order of 730 nm/815 nm, 660 nm/815 nm, and 660 nm/730 nm. This result was caused by the accuracy difference of the sensor at each band. The three-band vegetation indices TVI-1 and TVI-2 obtained by CGMD also showed a relatively high accuracy, with R^2^ values of 0.78 and 0.69, respectively, and RRMSE values of 0.10 and 0.27, respectively. The correlation of fitting of CGMD with FieldSpec HandHeld2 was relatively high, the R^2^ values were above 0.5 (except for DVI (R_660_, R_730_)), and CGMD could be used to obtain wheat canopy reflectance.

### 3.2. Fitting Results of Vegetation Indices With Growth Indices

#### 3.2.1. Fitting Results of Vegetation Indices with LAI

LAI refers to the total area of plant leaves per unit of land area, which is related to the size and structure of the plant canopy and is a comprehensive index reflecting the growth status of plants [31]. The fitting results of vegetation indices obtained by CGMD with LAI are shown in Figure 6. The number of samples used for the regressions was 63. Comparing the fitting results of each vegetation index with LAI, the correlation of fitting of the three-band vegetation indices TVI-1 and TVI-2 with LAI was high, with R^2^ values of 0.71 and 0.83, respectively, and RRMSE values of 0.33 and 0.24, respectively. Among them, the retrieval accuracy of TVI-2 on LAI was higher than those of most two-band vegetation indices. The fitting results of the two-band vegetation indices of different band combinations with LAI were different. The correlation of fitting of 730 nm/815 nm combination with LAI was the highest, and the R^2^ values were above 0.74. Among them, the fitting of DVI (R_730_, R_815_) was the best (R^2^ = 0.83, RRMSE = 0.23). The correlation of fitting of the 660 nm/815 nm combination was generally lower than that of the 730 nm/815 nm combination, and the fitting of DVI (R_660_, R_815_) was good (R^2^ = 0.81, RRMSE = 0.25); the 660 nm/730 nm combination showed the lowest correlation of fitting with LAI, the R^2^ values were lower than 0.68, and the RRMSE values were relatively high. In addition, the retrieval results of different types of vegetation indices exhibited differences. The fitting of DVI with LAI was better than that of NDVI and RVI in general (R^2^ > 0.68, RRMSE < 0.34).

The LAI prediction model was verified, and the verification result is shown in Figure 7. The prediction results of the two-band vegetation indices of the combinations of 660 nm/815 nm and 730 nm/815 nm and the three-band vegetation indices were relatively good. Among them, the prediction accuracy of DVI (R_660_, R_815_), DVI (R_730_, R_815_), and TVI-2 were higher than those of other vegetation indices; the corresponding R^2^ values were 0.64, 0.62, and 0.64, respectively, and the RRMSE values were 0.29, 0.30, and 0.29, respectively. The red spectral band is a strong chlorophyll absorption band, and its reflectance can reflect the LAI information. The DVI (R_660_, R_815_) and TVI-2 including the red spectral band improved the prediction accuracy of LAI.

The predicted value of each monitoring model deviated from the 1:1 line when the LAI value was large, and the prediction accuracy was low. The fitting results of vegetation indices and LAI showed (Figure 6) that the scattered point distribution of the vegetation indices was close to a straight line when the LAI was less than 3. As the LAI value increased, the magnitude of increase in vegetation index was less than that of LAI, and there was a saturation phenomenon, which reduced the monitoring accuracy of the monitoring model when the LAI value was large.

#### 3.2.2. Fitting Results of Vegetation Indices with LDW

LDW refers to the dry weight of leaves per unit area and is an important index of plant growth. The fitting results of vegetation indices obtained by CGMD with LDW are shown in Figure 8. The number of samples used for the regressions was 63. The experimental results showed that the fitting accuracy of the vegetation indices constructed in this paper with LDW were relatively high (R^2^ > 0.65). The correlation of fitting of the three-band vegetation indices TVI-1 and TVI-2 with LDW was higher than that of most two-band vegetation indices; R^2^ values were 0.77 and 0.78, respectively, and the RRMSE values were 0.30 and 0.27, respectively. Comparing the retrieval results of LDW using the two-band vegetation indices, the 660 nm/815 nm band combination and the 730 nm/815 nm band combination showed good fitting results with LDW; R^2^ values were above 0.72. Among them, the fitting result of DVI (R_660_, R_815_) with LDW was the best; the R^2^ value was 0.78, and the RRMSE value was 0.26. The red spectral band effectively improves the accuracy of CGMD to retrieve LDW.

The verification results of the LDW monitoring model are shown in Figure 9. The number of samples used for the verification was 28. The three-band vegetation indices and the two-band vegetation indices of 660 nm/815 nm and 730 nm/815 nm combinations showed relatively high prediction accuracy for LDW, with all R^2^ values being above 0.71 and RRMSE values being less than 0.32. Among them, the prediction accuracy of TVI-2 was the best; the R^2^ value reached 0.84, and the RRMSE value was 0.26. The LDW index reflects the number of leaves in a unit area. The red spectral band also contains the LDW information, and the addition of TVI-1 in the red spectral band effectively improves the monitoring accuracy of LDW. The wheat LDW monitoring model also produced saturation under the large value condition. The verification results showed that the monitoring model had a low prediction value when the LDW value was greater than 200 g/m^2^ and the accuracy of the LDW prediction for large groups needs to be improved.

#### 3.2.3. Fitting Results of Vegetation Indices with LNC

LNC is an important nutrient index reflecting crop growth, and obtaining crop LNC in real time can help guide variable-rate fertilization in the field. The fitting results of vegetation index obtained by CGMD with LNC are shown in Figure 10. The number of samples used for the regressions was 63. The correlation of fitting of vegetation indices with LNC was lower than those of other growth indices. The fitting results of three-band vegetation indices TVI-1 and TVI-2 with LNC did not improve the retrieval accuracy; the corresponding R^2^ values were 0.62 and 0.55, respectively, and the RMSE values were 0.15 and 0.17, respectively. Comparing the correlation of fitting of the two-band vegetation indices with LNC, the 660 nm/730 nm band combination had a poor fit with LNC, with R^2^ values of below 0.49, which was not suitable for LNC monitoring. The fitting results of the vegetation indices of 660 nm/815 nm and 730 nm/815 nm band combinations with LNC were relatively good; the range of R^2^ was 0.53–0.64, and the range of RRMSE was 0.15–0.17. Among them, the correlation of fitting of NDVI (R_730_, R_815_) with LNC was the highest (R^2^ = 0.64, RRMSE = 0.15).

The verification results of the LNC prediction model are shown in Figure 11. The number of samples used for the verification was 28. The three-band vegetation indices and the two-band vegetation indices of 660 nm/730 nm and 660 nm/815 nm combinations showed low prediction accuracy for LNC. All R^2^ values were below 0.5, the RRMSE values were large, and the trend line deviated from the 1:1 line, so they were not suitable for LNC prediction and measurement. The 730 nm/815 nm combination had relatively high prediction accuracy for LNC. Among them, the prediction accuracy of NDVI (R_730_, R_815_) for LNC was the highest, with R^2^ of 0.60 and RRMSE of 0.17. There was no saturation problem in the wheat LNC monitoring model, but the dispersion of scattered points of the model was relatively high, and the prediction accuracy was lower than those of other vegetation indices. When the LNC was lower than around 3%, the predicted value was higher than the measured value; when it was greater than around 3%, the predicted value was lower than the measured value, and the fitted curve deviated from the 1:1 line. The red spectral band is the main absorption band of chlorophyll and can indirectly reflect the nitrogen status of leaves in theory. However, the results in this paper demonstrate that the red spectral band does not improve the retrieval accuracy of LNC, and further research on LNC monitoring is needed.

#### 3.2.4. Fitting Results of Vegetation Indices with LNA

LNA is a growth index obtained through the calculation using LDW and LNC, and it reflects both the nitrogen status of crop leaves and the growth characteristics of crop population. The fitting results of the vegetation indices obtained by CGMD with LNA are shown in Figure 12. The number of samples used for the regressions was 63. The experimental results showed that the correlation of fitting of three-band vegetation indices TVI-1 and TVI-2 with LNA was high. The R^2^ values were 0.83 and 0.85, respectively, but the model training error was relatively large, and the RRMSE values were 0.37 and 0.32, respectively. Comparing the results of the two-band vegetation indices on the retrieval of LNA, the fitting results of the 660 nm/815 nm and 730 nm/815 nm band combinations were better than that of the 660 nm/730 nm combination, with R^2^ values being above 0.73. Among them, NDVI (R_730_, R_815_), RVI (R_730_, R_815_), DVI (R_660_, R_815_), and DVI (R_730_, R_815_) showed good fitting results with LNA, with R^2^ values of 0.83, 0.82, 0.85, and 0.84, respectively, and RRMSE values of 0.36, 0.37, 0.33, and 0.31, respectively.

The verification results of the LNA monitoring model are shown in Figure 13. The number of samples used for the verification was 28. The accuracies of the three-band vegetation indices and the two-band vegetation indices of 730 nm/815 nm for the prediction of LNA were relatively high, and R^2^ values were above 0.73. However, some monitoring models had large errors, and the RRMSE range was 0.30–0.51. Among them, the prediction accuracy of NDVI (R_730_, R_815_) was the best, with an R^2^ of 0.82 and an RRMSE of 0.30. LNA is a growth index obtained by multiplying LDW and LNC. The fitted curve showed a similar saturation phenomenon as that of LDW. The prediction accuracy of the monitoring model decreased when the LNA value was greater than 10 g/m^2^. The experimental results show that the vegetation indices of the combination of red edge and near-infrared band are more suitable for the prediction of wheat LNA.

### 3.3. Spectral Monitoring Model of Wheat Growth

From the fitting of the vegetation index with the growth index, under the condition that the wheat canopy population was large, the vegetation index was saturated, which led to poor prediction accuracy of each vegetation index with high biomass. As shown in Figure 6, Figure 8 and Figure 10, as the value of growth indices increased, the distribution of scattered points became more discrete, and this problem was difficult to solve with a single spectral monitoring model. According to the pattern of wheat growth, various growth indices in the vegetative growth stage tended to increase rapidly, and after the shift from vegetative growth to reproductive growth, the trend of increase was flat, and some indices even decreased. Based on the differences of wheat growth indices in different phenological periods, the samples could be classified, which could reduce the gap between the values of scattered points with high biomass in each model. Therefore, this paper proposes a method for constructing a wheat-growth-index spectral monitoring model based on the growth stages. The data were divided into periods I and II according to the characteristics of wheat growth, and a growth spectral monitoring model was independently constructed to solve the problem of impact of vegetation index saturation on the prediction accuracy of the model.

The dynamic changes of wheat growth indices from the jointing stage to the flowering stage are shown in Figure 14. The value of LAI before the heading stage increased over the growth period; the increase rate from the jointing stage to the booting stage was relatively small, the average value increased from 1.75 to 2.06, and the value ranges were 0.55–2.87 and 0.69–3.88, respectively. The value from the booting stage to the heading stage varied greatly; the average value increased to 2.80, and the value at the heading stage ranged from 0.80 to 5.48. After the heading stage, the wheat LAI value changed slightly, the average value at the flowering stage was 2.98, and the value range was 1.02–5.54. Therefore, the period from the jointing stage to the booting stage was defined as period I, and the period from the heading stage to the flowering stage was defined as period II.

The LDW value of wheat increased greatly from the jointing stage to the booting stage; the average value increased from 112.13 g/m^2^ to 191.48 g/m^2^, and the value range increased from 36.56–199.52 g/m^2^ to 53.35–371.49 g/m^2^. The value from the booting stage to the flowering stage was relatively stable, the value from the booting stage to the heading stage decreased, and the average value decreased from 191.48 g/m^2^ to 144.05 g/m^2^. The value range at the booting stage was 27.83–302.18 g/m^2^. The value from the booting stage to the flowering stage remained the same. The average value at the flowering stage was 143.03 g/m^2^, with a range of 46.64–330.31 g/m^2^. Therefore, the jointing stage alone was defined as period I, and the booting stage to the flowering stage was defined as period II.

The differences of LNC in the wheat phenological stages were small, and the average values from the jointing stage to the flowering stage were 2.93%, 2.77%, 2.82%, and 2.84%, respectively. The range of the jointing stage was close to that of the booting stage, and the value range was mostly distributed at 1.7%–3.5%, while the value ranges of the heading stage and the flowering stage were more consistent, and the value range was mostly distributed at 1.8%–4.0%. Therefore, the jointing stage and the booting stage were defined as period I, and the heading stage and flowering stage were defined as period II.

LNA is a growth index obtained by multiplying LNC and LDW. Its trend of change includes the characteristics of LDW and LNC. There was a large change in the value from the jointing stage to the booting stage, the average value increased from 3.45 g/m^2^ to 5.70 g/m^2^, and the range increased from 0.64–7.13 g/m^2^ to 0.94–13.27 g/m^2^. The values from the booting stage to the flowering stage tended to be stable. The average values were 5.70 g/m^2^, 4.47 g/m^2^, and 4.51 g/m^2^, respectively, and the values ranged from 0.94–13.27 g/m^2^, 0.55–10.00 g/m^2^, and 1.03–11.97 g/m^2^, respectively. Therefore, the jointing stage alone was defined as period I, and the period from the booting stage to the flowering stage was defined as period II.

The vegetation indices with the highest prediction accuracy corresponding to different wheat growth indices of periods I and II were screened out to construct the crop growth monitoring model (Figure 15). The number of samples used for constructing models and the verification was 63 and 28, respectively. The vegetation index corresponding to LAI with the highest prediction accuracy was TVI-2, the vegetation index corresponding to LDW was TVI-1, the vegetation index corresponding to LNC was NDVI (R_730_, R_815_), and the vegetation index corresponding to LNA was NDVI (R_730_, R_815_). The monitoring model and verification results are shown in Figure 15. The fitting accuracies of LAI, LDW, and LNA in period II increased, R^2^ values increased to 0.94, 0.85, and 0.87, respectively, and the RRMSE values were 0.13, 0.08, and 0.32, respectively, showing that the saturation problem caused by large groups in the late growth stage was eliminated to some extent. However, the fitting result of period I was not considerably improved compared with that of the original monitoring model, and the quality of fitting results of the growth indices, except for LAI and LNC, decreased to a certain extent. The reason may be that the data range and coefficient of variation of the early growth stage are small, and under the same conditions, the required accuracy of the monitoring instrument is higher. Additionally, the reduction in the number of training samples for modeling in different periods also has a certain impact on the accuracy of the model. Therefore, in this study, we still chose the monitoring model of the whole growth period when predicting the values of the growth indices LDW and LNA in period I. The trend lines, R^2^ values, and RRMSE values of LNC in period I and period II were consistent, indicating that the change in the numerical characteristics of LNC in different phenological periods is small, and segmentation modeling cannot improve the accuracy of LNC fitting.

The verification results showed that the prediction accuracy of the LAI monitoring model increased the most, R^2^ increased from 0.64 to 0.79, and RRMSE decreased from 0.29 to 0.22. The prediction accuracy improved to a certain extent when the population was large, and the trend line was closer to the 1:1 line. The prediction accuracies of LDW and LNA improved slightly, the R^2^ values increased to 0.85 and 0.85, respectively, and the RRMSE values were 0.23 and 0.28, respectively. The prediction result of LNC was the same as that of the original monitoring model, and the segmentation modeling could not improve its monitoring accuracy. The above finding shows that the wheat growth monitoring model constructed based on the growth stages in this study greatly improves the monitoring accuracy of LAI, but the prediction accuracy of LDW, LNC, and LNA were not able to be substantially improved.

## 4. Discussion

With the development of non-destructive monitoring technologies based on feature identification through reflectance spectra, some research institutes in China and abroad have developed a variety of crop growth monitoring apparatuses [32,33]. Compared with GreenSeeker and other two-band spectrophotometers, CGMD can obtain more vegetation indices, and it can choose the corresponding vegetation index to monitor different growth indices. Although ASD FieldSpec and other hyperspectral spectrophotometers can obtain more vegetation indices, they are difficult to popularize because of their large data redundancy and complex data processing. The spectral bands of CGMD are sensitive to crop growth indices, which makes it easier to operate and more suitable for crop field monitoring.

The addition of a 660 nm band sensor to CGMD increased the degree of freedom of vegetation index selection, and the three-band vegetation indices constructed in this paper improved the prediction accuracy of the growth indices LAI and LDW. The red spectral region is a strong chlorophyll absorption band, and its change in reflectance change can indicate the area of green plants in the field-of-view to a certain extent. Previous studies have shown that the red and near-infrared bands are more sensitive to canopy population indices such as LAI and LDW, which is consistent with the results of this paper [34,35]. Therefore, the addition of the 660 nm band sensor to CGMD is an effective supplement to the original 730 nm and 815 nm band sensors of CGMD-302. The correlations of fitting the two-band vegetation indices based on the combinations of 660–730 nm and 660–815 nm bands with each growth index were always lower than those of the 730–815 nm combination. This result may be caused by the spectral characteristics of red light. The red spectral band is the absorption peak of plant chlorophyll, and the reflectance value is low. Therefore, the red spectral band requires higher accuracy of the sensor than the red edge and near-infrared bands. The results of this study also showed that the accuracy of the reflectance obtained at the 660 nm band was lower than those at other bands, which limits the ability of the red spectral band to retrieve the growth index to a certain extent. Conversely, the red spectral band is more prone to saturation when monitoring various growth indices. Yao et al. [36] found that the NDVI became saturated when biomass reached about 400 g/m^2^ or when plant N uptake reached about 10 g/m^2^. Wang et al. [37] found that when LAI was less than 3, the red band was more sensitive to LAI, while the green band was more sensitive when LAI reached 3. Previously, researchers have alleviated this issue to some extent by creating a new vegetation index, but there is still no accepted method to eliminate the saturation phenomenon [38,39,40]. Therefore, although red is a sensitive band of many growth indices, there are still many problems to be solved in application.

The canopy reflectance and structure of wheat were affected by nutrition and growth period. The great difference of growth indices from jointing stage to flowering stage would decrease the precision accuracy of monitoring model [41,42,43]. Li et al. [44] found that the canopy structure of winter wheat had obvious difference between feek 4–7 and feek 8–10, and the prediction accuracy of nitrogen concentration could be effectively increased through building monitoring models independently at different growth stages. Yu et al. [45] indicated that growth stages negatively affect hyperspectral indices in different ways in nitrogen monitoring of rice, and the sensitive band and vegetation index were different between the before and after heading stages. Zhao et al. [46] found that the monitoring accuracy of winter wheat LAI increased by dividing the growth period of winter wheat into three stages and building the monitoring model independently. Li et al. [47] indicated that LAI = 3 was a suitable segmentation value, and the segmented winter wheat monitoring model could improve the prediction accuracy and alleviate the saturation. A crop growth monitoring model was constructed for the two periods separately, which effectively improved the prediction accuracy of LAI, but did not considerably enhance those of LDW, LNC, and LNA. Over the growth period, the gap between high-nitrogen treatment and low-nitrogen treatment on biomass further increased, and the coefficient of variation increased. Staged modeling will still be affected by saturation during the late growth stage. It should be pointed out that the population size of wheat treated with different nitrogen treatments is greatly different, and the construction of monitoring models for different nitrogen treatments can theoretically improve the prediction accuracy. However, this method needs the information of the nitrogen application level in the area to be tested in advance, and this is complex in practical operation, so it was not adopted in this study. In addition, LNC is mostly affected by nitrogen level and changes slightly with phenological period. The results of this study show that the modeling of multiple growth periods is not suitable for improving the accuracy of LNC monitoring.

The ability of CGMD to monitor wheat LNC was always lower than that of the other growth indices, and the vegetation indices of all combinations of red, red edge, and near-infrared bands did not effectively improve the monitoring accuracy. Verstraete [48] showed that the best vegetation index should be more sensitive to the target indices and less sensitive to other indices. The wavebands used in this paper are similar to the vegetation indices that they constructed and are highly sensitive to the canopy population size index; they will inevitably be affected by the canopy population size when performing LNC monitoring. Xue et al. [49] demonstrated that the ratio constructed from a combination of the green and blue spectral bands and the normalized vegetation index showed a significant negative correlation with the nitrogen content of rice leaves, and the prediction accuracy reached 80.09%. Tian et al. [50,51] used blue and green spectral bands to monitor rice and found that the R^2^ of the new green band vegetation index SR (545, 538) in prediction of wheat LNC reached 0.73, while the new index showed a low sensitivity to LAI, which eliminated the influence of canopy size on the prediction of wheat LNC to a certain extent. The R^2^ of three-band vegetation index R_434/_(R_496_ + R_401_) constructed by the blue spectral band in prediction of rice LNC reached 0.84, and the universality was also good. Therefore, the selection of sensors with more wavelength combinations such as green and blue bands can be considered in the development of multi-spectral crop growth sensors in the future.

## 5. Conclusions

(1) A portable CGMD three-band instrument was used to obtain wheat canopy vegetation indices at the jointing, booting, heading, and flowering stages and fitted them with a commercially available instrument by ASD FieldSpec HandHeld2 spectroradiometer. The results showed that the CGMD instrument could effectively obtain the wheat canopy spectral data, and the measurement results were accurate and stable. Using CGMD we could effectively predict the LAI, LDW, LNC, and LNA of the wheat; the R^2^ values were 0.64, 0.84, 0.60, and 0.82, respectively, and the RRMSE values were 0.29, 0.26, 0.17, and 0.30, respectively. The addition of the red spectral band to CGMD, which was in addition to the existing near-infrared and red edge bands, effectively improved the monitoring of wheat LAI and LDW. CGMD has the advantages of high monitoring accuracy, multiple invertible growth indices, simple operation and high cost performance, which is suitable for agricultural production.

(2) Regarding the issue of vegetation index saturation, this study proposed a method for constructing a wheat-growth-index spectral monitoring model based on the growth stages. The spectral monitoring model based on the growth stages increased the prediction accuracy, of LAI, LDW, and LNA; the R^2^ values were 0.79, 0.85, and 0.85, respectively, and the RRMSE values were 0.22, 0.23, and 0.28, respectively. The method of constructing the growth parameter spectral monitoring model based on the growth stages of wheat effectively solves the problem of saturation of the vegetation indices in the late growth stage of wheat, and the CGMD reached a relatively high level of prediction on the growth indices LAI, LDW, and LNA.

## Figures and Tables

**Figure 1 sensors-20-02894-f001:**
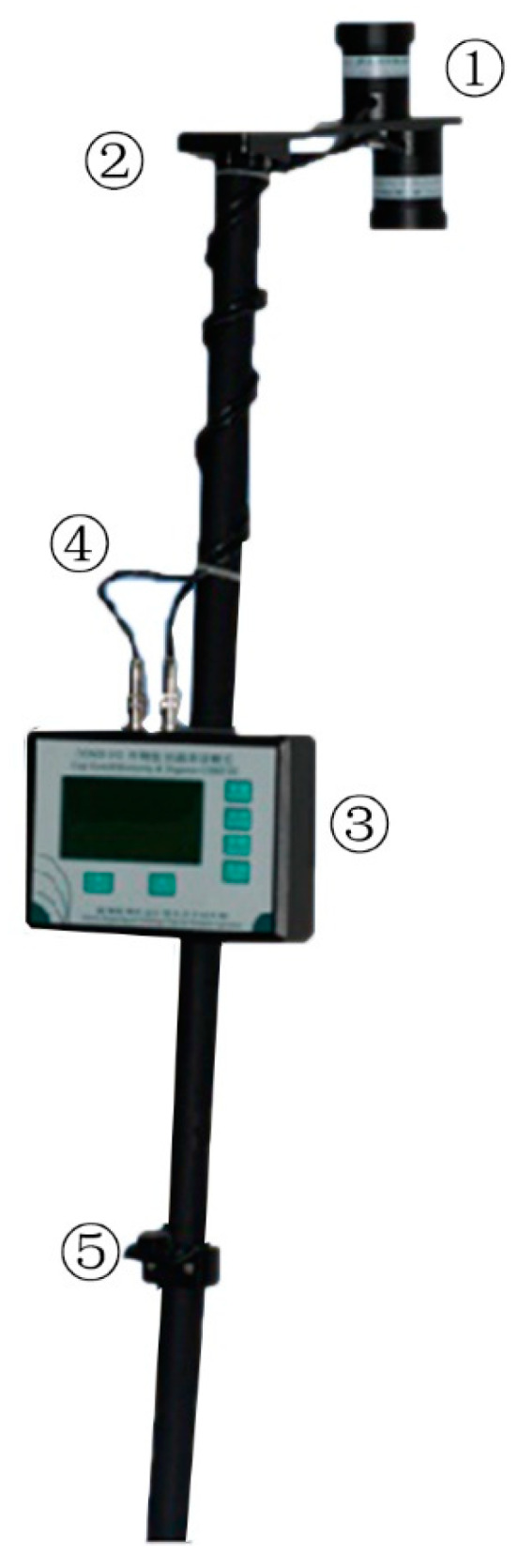
Portable three-band instrument of crop-growth monitoring and diagnosis: ① Multi-spectral crop sensor; ② Sensor support; ③ Processor system; ④ Shielded cable; ⑤ Level.

**Figure 2 sensors-20-02894-f002:**
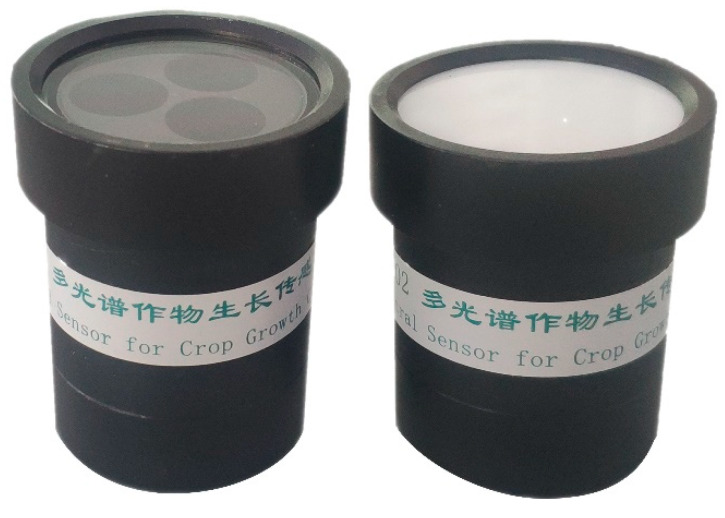
Multispectral sensor.

**Figure 3 sensors-20-02894-f003:**
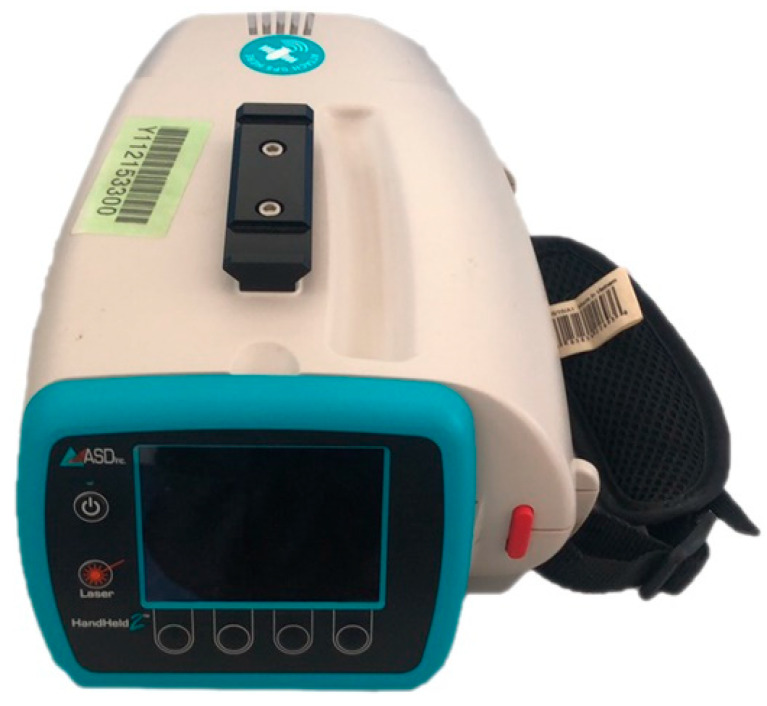
ASD FieldSpec HandHeld2 spectrophotometer.

**Figure 4 sensors-20-02894-f004:**
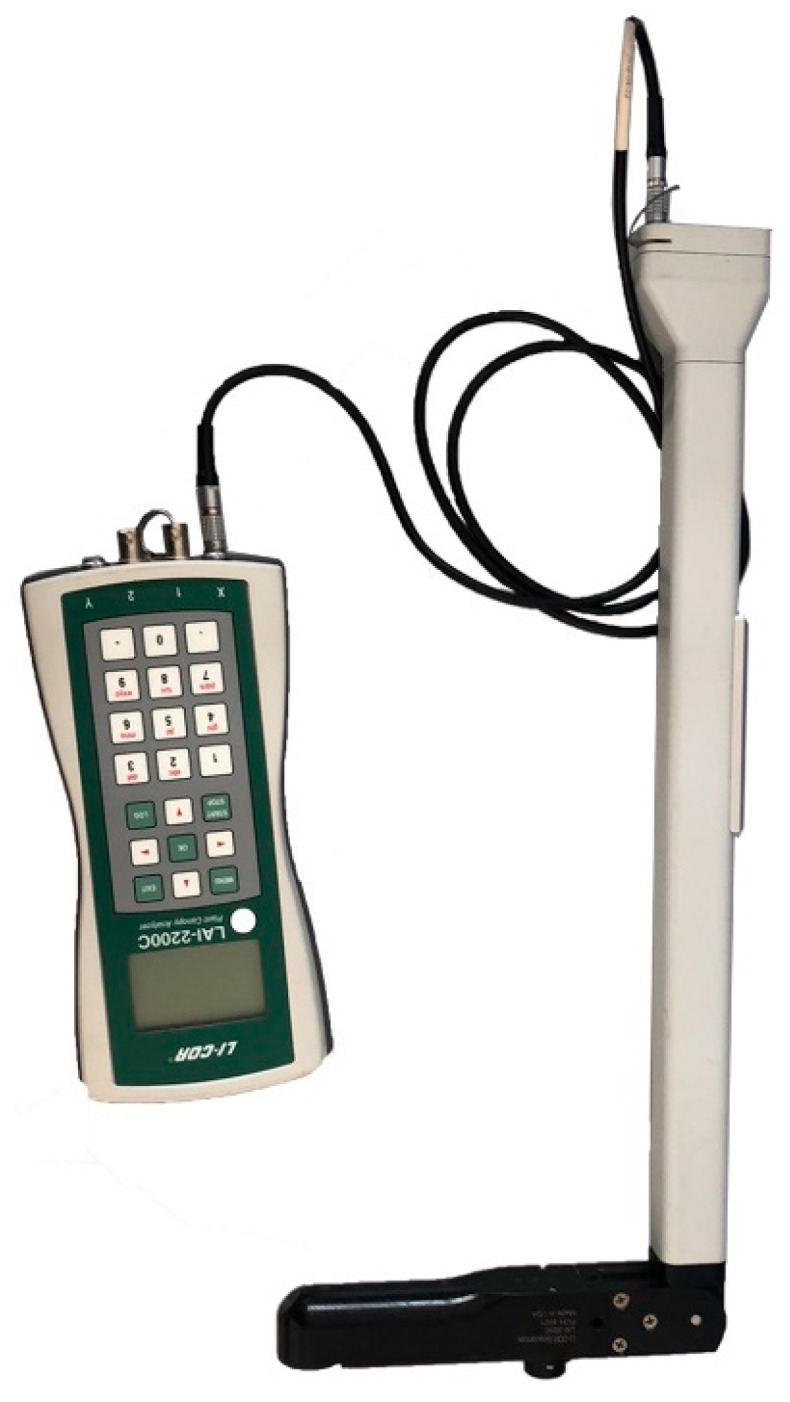
LAI-2200C plant canopy analyzer.

**Figure 5 sensors-20-02894-f005:**
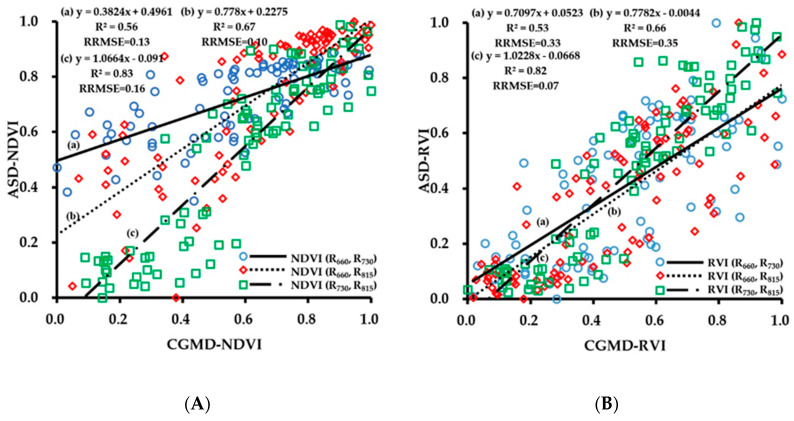
(**A**) Fitting curve of CGMD-NDVI and ASD-NDVI; (**B**) Fitting curve of CGMD-RVI and ASD-RVI; (**C**) Fitting curve of CGMD-DVI and ASD-DVI; (**D**) Fitting curve of CGMD-TVI and ASD-TVI.

**Figure 6 sensors-20-02894-f006:**
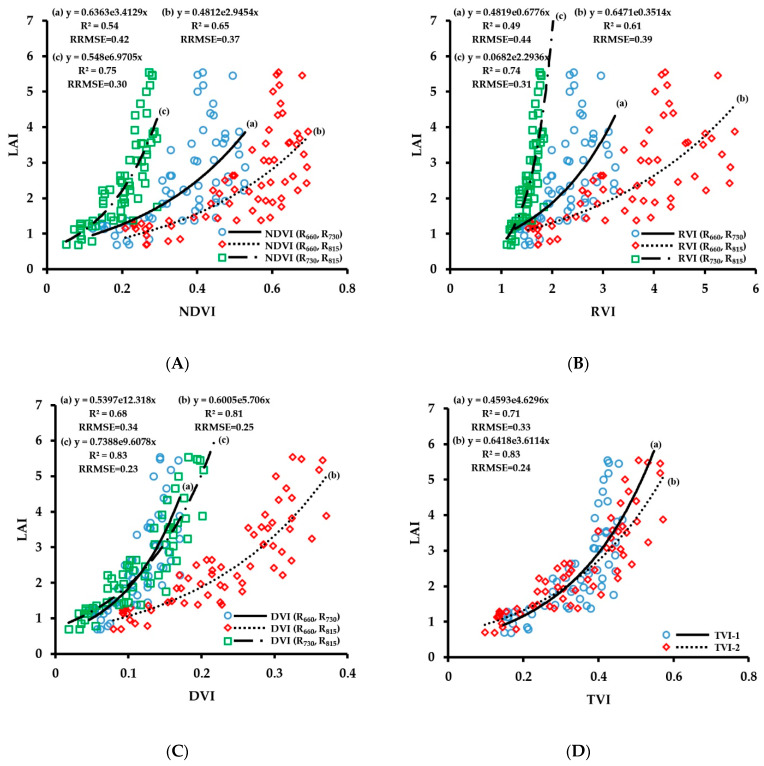
Fitting curve of NDVI (**A**), RVI (**B**), DVI (**C**) and TVI (**D**) to LAI of wheat.

**Figure 7 sensors-20-02894-f007:**
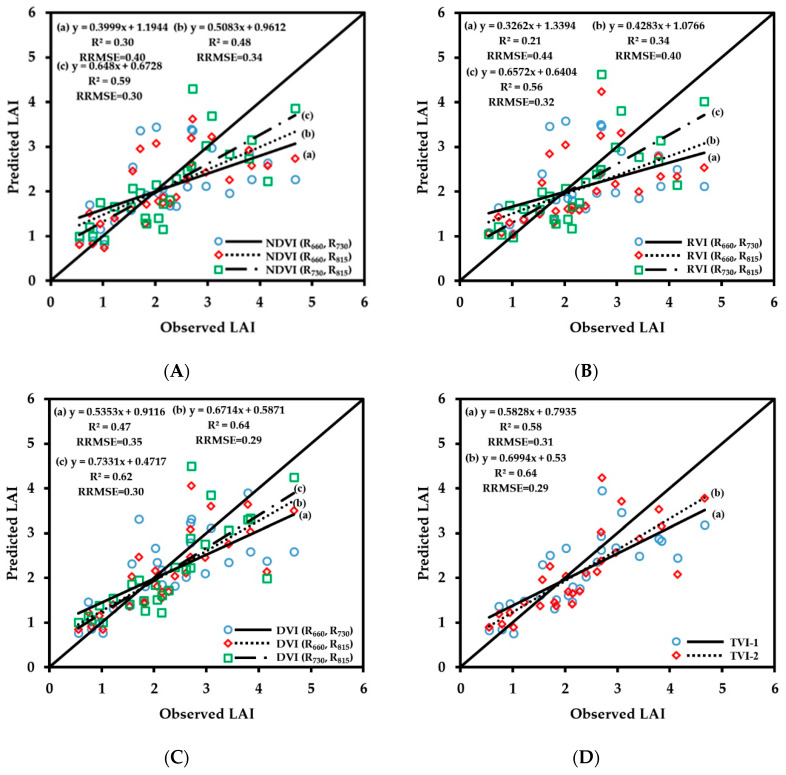
The relationships between observed and predicted LAI values of wheat varieties based on NDVI (**A**), RVI (**B**), DVI (**C**) and TVI (**D**) models.

**Figure 8 sensors-20-02894-f008:**
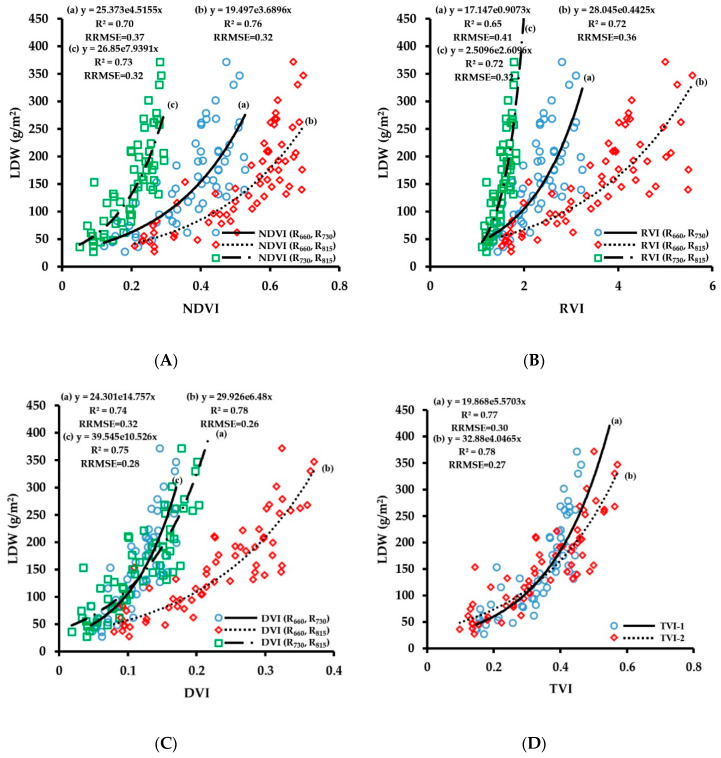
Fitting curve of NDVI (**A**), RVI (**B**), DVI (**C**) and TVI (**D**) to LDW of wheat.

**Figure 9 sensors-20-02894-f009:**
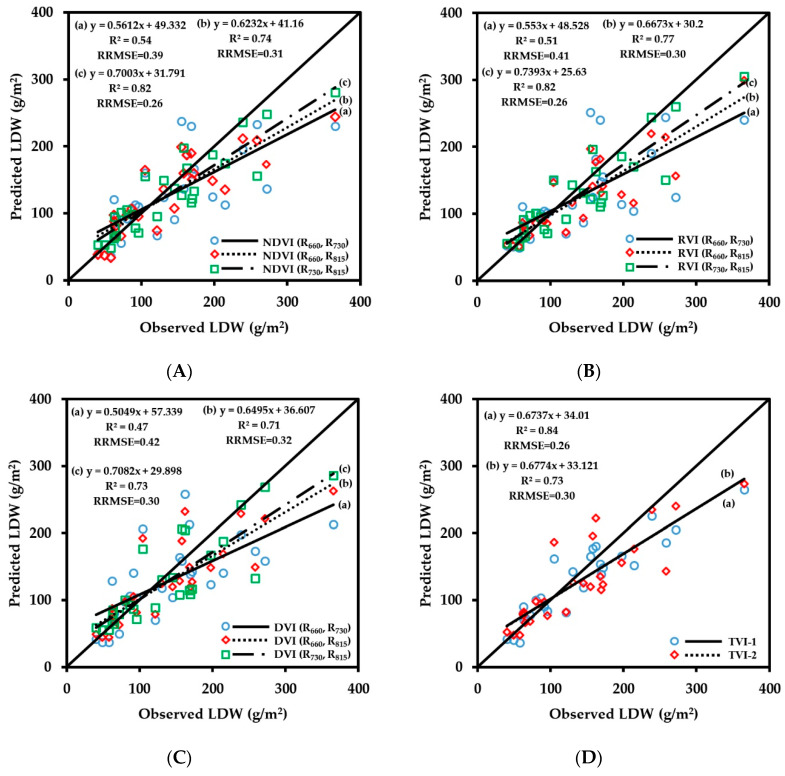
The relationships between observed and predicted LDW values of wheat varieties based on NDVI (**A**), RVI (**B**), DVI (**C**) and TVI (**D**) models.

**Figure 10 sensors-20-02894-f010:**
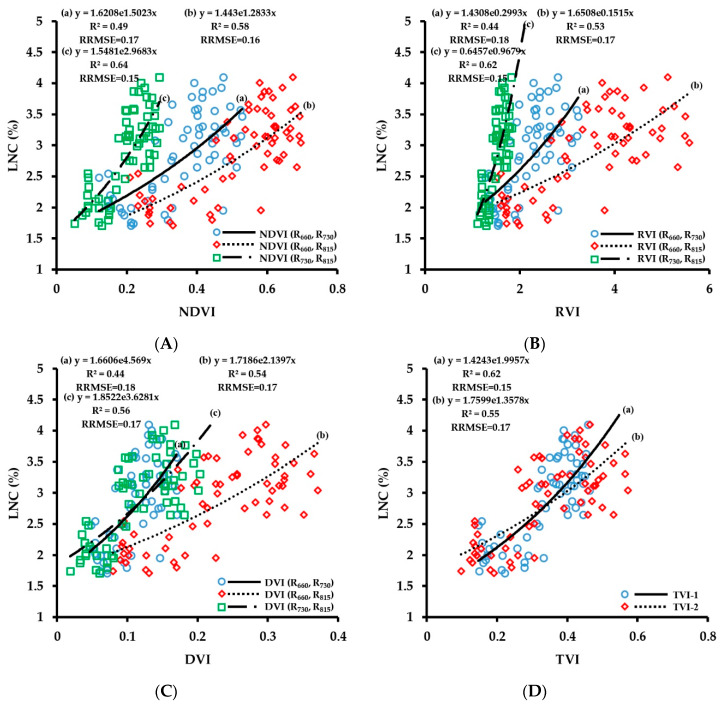
Fitting curve of NDVI (**A**), RVI (**B**), DVI (**C**) and TVI (**D**) to LNC of wheat.

**Figure 11 sensors-20-02894-f011:**
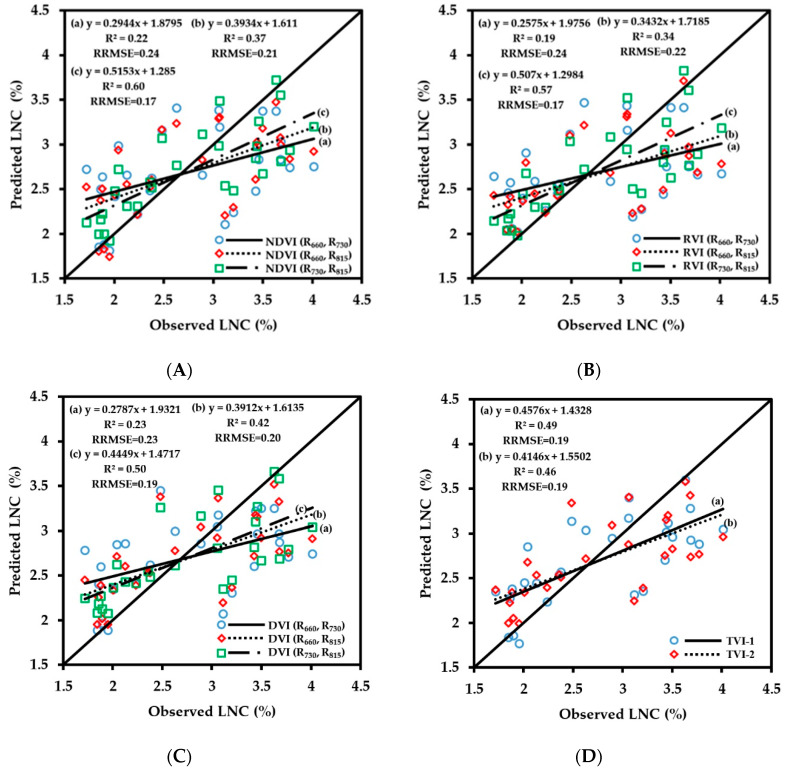
The relationships between observed and predicted LNC values of wheat varieties based on NDVI (**A**), RVI (**B**), DVI (**C**) and TVI (**D**) models.

**Figure 12 sensors-20-02894-f012:**
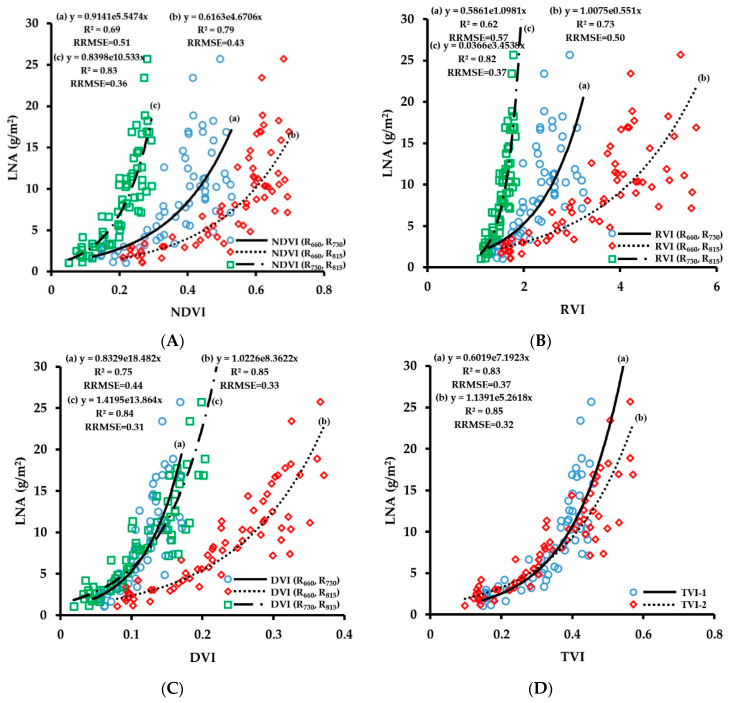
Fitting curve of NDVI (**A**), RVI (**B**), DVI (**C**) and TVI (**D**) to LNA of wheat.

**Figure 13 sensors-20-02894-f013:**
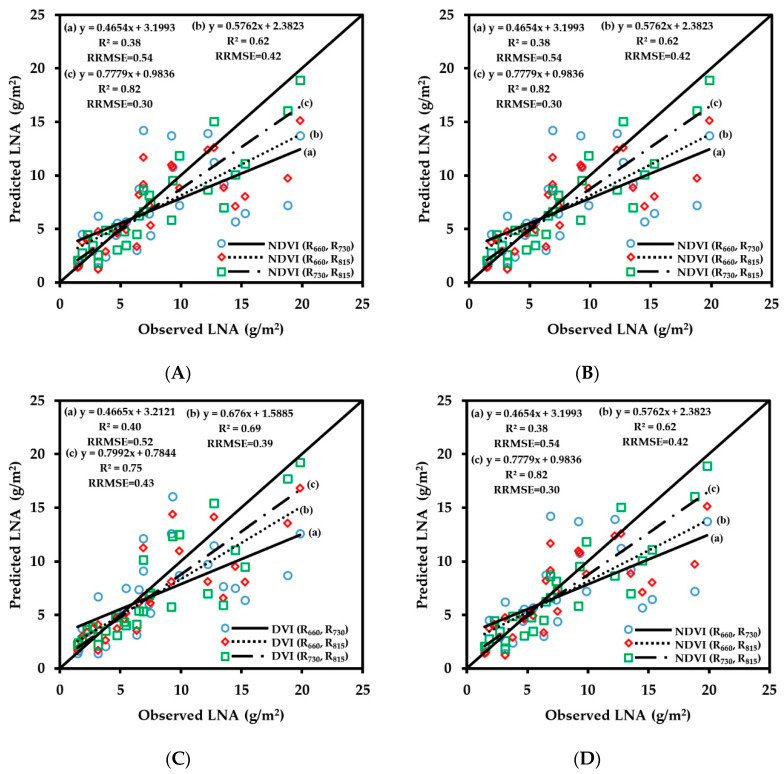
The relationships between observed and predicted LNA values of wheat varieties based on NDVI (**A**), RVI (**B**), DVI (**C**) and TVI (**D**) models.

**Figure 14 sensors-20-02894-f014:**
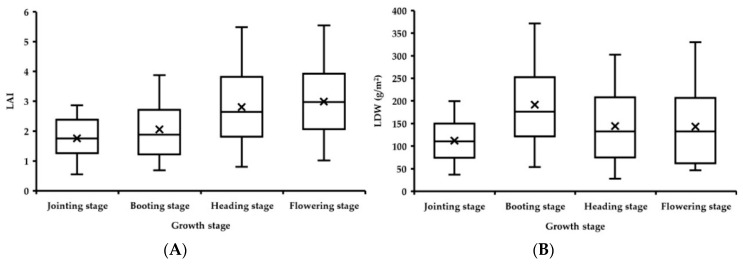
LAI (**A**), LDW (**B**), LNC (**C**) and LNA (**D**) of wheat at different growth stages.

**Figure 15 sensors-20-02894-f015:**
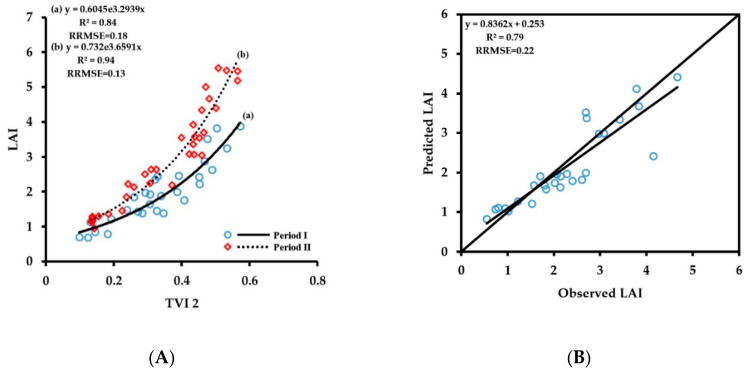
Fitting curves of vegetation index to LAI (**A**), LDW (**C**), LNC (**E**) and LNA (**G**) of wheat at periodI and periodII; The relationships between observed and predicted LAI (**B**), LDW (**D**), LNC (**F**) and LNA (**H**) values based on the multi-stage monitoring model of wheat.

**Table 1 sensors-20-02894-t001:** The parameters of some instruments of crop-growth monitoring.

Instrument	Sensor Size (mm)	Weight (kg)	Price (CNY)	Waveband (nm)	Light Source
CGMD	54 × 38 × 38	1.6	About 4000	660, 730, 815	Sunlight
ASD Handheld 2	215 × 140 × 90	1.2	Over 150,000	325–1075	Sunlight
Crop Circle ACS-470	201 × 89 × 48	3.6	Over 100,000	450, 550, 650, 670, 730, 760	LED
GreenSeeker Handheld	277 × 86 × 15	1.02	Over 100,000	656, 774	LED
SPAD-502	164 × 78 × 49	0.23	Over 10,000	650, 940	LED
Dualex 4	205 × 65 × 55	0.22	Over 100,000	375, 655, 710, 850	LED

CGMD: a portable three-band instrument for crop-growth monitoring and diagnosis.

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
