# Peer review of "Monitoring Wheat Growth Using a Portable Three-Band Instrument for Crop Growth Monitoring and Diagnosis"

_sensors, 2020, doi:10.3390/s20102894_

Round 1

Reviewer 1 Report

This paper constructed a novel three wavelength instrument for crop-growth monitoring and diagnosis. And authors proposed a model for growth monitoring. With the instrument and the model, LAI, LDW, LNC, and LNA were estimated with acceptable accuracy. The quality of this paper is good, and the experimental design is sound. The language should be improved. In addition, I would like to encourage authors to compare their system’s performance with other systems described in section of introduction. That would help for the evaluation of their system.

Line 52. “Chinese and international research institutions”, the word “institutes” seems to be right.

Line 102. Why authors chose the red band for the third band? Why green band is not used?

Line 111. More researches on the wheat growth monitoring models should be included in the section of introduction.

Line 130. The details of the three-band multispectral sensor should be described. Is this multispectral sensor bought or constructed by the authors? How did it realize the three-band detection?

Line 143. Why ASD FieldSpec HandHeld2 was used should be explained here. The explanation should also be described for LAI-2200C.

Line 205. The model used for LAI, LDW, LNC, and LNA estimation is the regression analysis? Authors should clarify the models for different stages.

Author Response

Point 1. The language should be improved.

Response 1:

The language of the manuscript has been improved at LetPub. Their language editors are native English speakers with long-term experience in editing scientific and technical manuscripts. Some language problems have been revised.

Point 2. I would like to encourage authors to compare their system’s performance with other systems described in section of introduction.

Response 2:

Our system’s performance has been compared with other systems.

Point 3. Line 52. “Chinese and international research institutions”, the word “institutes” seems to be right.

Response 3:

Line 52. “Chinese and international research institutions” has been replaced with “Chinese and international research institutes”.

Point 4. Line 102. Why authors chose the red band for the third band? Why green band is not used?

Response 4:

The reason for choosing the red band has been added to the introduction section.

Point 5. Line 111. More researches on the wheat growth monitoring models should be included in the section of introduction.

Response5:

More researches on the wheat growth monitoring models have been added in the section of Introduction. Some researches on the models for different stages have also been added to the section of Discussion.

Point 6. Line 130. The details of the three-band multispectral sensor should be described. Is this multispectral sensor bought or constructed by the authors? How did it realize the three-band detection?

Response 6:

The details of the three-band multispectral sensor have been described. The multispectral sensor was conducted by the authors. The principle of the three-band detection of this sensor has also been added.

Point 7. Line 143. Why ASD FieldSpec HandHeld2 was used should be explained here. The explanation should also be described for LAI-2200C.

Response 7:

ASD FieldSpec HandHeld2 was used to evaluate the data acquisition performance of the portable three-band CGMD instrument. LAI-2200C was used to acquire observed LAI. The purpose of these two apparatuses has been added to the corresponding section.

Point 8. Line 205. The model used for LAI, LDW, LNC, and LNA estimation is the regression analysis? Authors should clarify the models for different stages.

Response 8:

Regression analysis was used to construct the model for LAI, LDW, LNC, and LNA estimation. Related instructions have been added to the Materials and Methods section.

Reviewer 2 Report

The work deals with a very interesting topic. However, in my opinion the authors present preliminary results and do not justify them. There is no relationship between the design of the experiment and the results presented. Discussion of the results is sparse. There are very few bibliographic references. No references on analytical methods. The conclusions are results.

Author Response

Point 1. The authors present preliminary results and do not justify them.

Response 1:

We agree with you that our result needs to be verified. The prediction accuracy of our monitoring models was verified in the section of Results. For example, the fitting results of Fig.15 (B) verified the prediction accuracy of the model constructed in Figure 15 (A). The data used in Figure 15(A) and Figure 15(B) were from different plots and were independent of each other. More verification tests will be carried out at different ecological sites for different varieties of wheat in future.

Point 2. There is no relationship between the design of the experiment and the results presented.

Response 2:

The relationship between our design and the results reflected the following two aspects.

  • We designed three nitrogen fertilizer levels and three planting densities in the experiment to obtain wheat plots with diverse canopy structure to evaluate our instrument. This is a commonly used experiment design method in wheat cultivation. Diverse spectral reflectance and growth indices were obtained at different nitrogen fertilizer levels and planting densities. On this basis, we comprehensively evaluated the performance of CGMD.
  • Spectral data and growth indices were obtained simultaneously as four of the growth stages of wheat: jointing, booting, heading, and flowering. The segment basis of different growth stages in our research was the phenological characteristics of wheat population. Based on the data obtained at different stages, we proposed the method of model construction.

The section of Materials and Methods has also been revised.

Point 3. Discussion of the results is sparse.

Response 3:

Some content has been added to the Discussion section. We compared CGMD with other commercial tools, and increased the discussion of monitoring models at different stages. We hope that these additional contents can enrich the Discussion section.

Point 4. There are very few bibliographic references. No references on analytical methods.

Response 4:

More references have been added to this manuscript, especially on analytical methods.

Point 5. The conclusions are results.

Response 5:

The Conclusion section has been revised.

Reviewer 3 Report

The paper is timely. However, the related work on the subject topic has been addressed properly.

Authors are advised to address related work on internet of underground things, and smart agriculture enabling technologies.

Figure quality can be improved as well.

Moreover, advantage of the system as compared to existing technlogies should be highlighted as well.

Author Response

Point 1. The related work on the subject topic has been addressed properly. Authors are advised to address related work on internet of underground things, and smart agriculture enabling technologies.

Response 1:

Although there have been many researches on portable instrument for crop growth monitoring, our research has the following advantages.

  • CGMD can simultaneously obtain the reflectance of red, red edge, and near-infrared spectral bands. According to the modeling needs of the growth indices of different crops, the appropriate vegetation index was then able to be selected, which allows for more flexibility and applicability in practical applications than two-band instruments.
  • Compared with hyperspectral instruments, CGMD has no problem of data redundancy and is easy to operate. People without professional knowledge can also operate CGMD well.
  • The main objects of commercial instruments are still research institutes at present. the high cost of these instrumental fail to meet the requirements of precision agriculture for acquiring field information at low cost. CGMD exhibits a simple structure, high integration, and good cost performance, and therefore is more suitable for application in crop field production.
  • We proposed a method to construct the wheat-growth-index spectral monitoring models that were defined according to the growth periods. This method is easy to operate and can alleviate the saturation of growth indices such as LAI.

We agree with you about the internet of underground things, and smart agriculture enabling technologies. We will consider to address related work in the future.

Point 2. Figure quality can be improved as well.

Response 2:

Figure quality has been improved. Some unclear pictures have been remade.

Point 3. Advantage of the system as compared to existing technologies should be highlighted as well.

Response 3:

Advantages of our system as compared to existing technologies have been added to the section of Discussion.

Round 2

Reviewer 1 Report

  1. Line 18: the existing two-band instrument: the single form is used here, do you indicate comparison with a single two-band instrument? If so, please provide its name and relevant wavelengths.
  2. Line 19: “relying on a single vegetation index”: this is incorrect. Even with only two wavelengths, different kinds of vegetation indices can be computed, eg. NDVI, DVI…
  3. Line 19: “low accuracy of growth index retrieval”: this is inaccurate. I’m sure the performance is not always poor, please refine your statement.
  4. Line 27: “could be used for monitoring the canopy vegetation index” index or indices, please make sure.
  5. Line 29: 0.64, 0.84 … 0.82, respectively.
  6. Line 29, LAI, LDW, LNC, LNA: these abbreviations were not explained in their first appearance.
  7. Line 35: do you mean that the RRMSE was improved by 0.22, 0.23 …? Note that the verb of the whole sentence was “improved”.
  8. Line 48, “they require destructive sampling are tedious, time-consuming”: with two verbs, this sentence doesn’t make sense.
  9. I’m surprised that the whole paragraph from line 52 to 65 said nothing about any airborne or satellite multispectral or hyperspectral sensors. If the authors want to confine their introduction to only portable sensors, state this clearly in the first place to avoid any misunderstanding.
  10. Line 89-90, what do you mean by saying “only output the vegetation index”? Again, even for a two-band sensor, various kinds of indices can be computed. Also, as the authors suggested that the above-mentioned spectrometers were “costly, complicated to operate”, their approximate price, size, weight, as well as measurement manners etc. in comparison with the adopted system in this study should be clearly listed, emphasizing the importance of this new system (maybe in section 2.2). As far as I’m concerned, some of the mentioned systems (e.g. ASD) are not “complicated to operate”.
  11. Line 113-114, the authors said that two bands is not enough. Given so, why was only three bands adopted in this study? Why is three the optimal number, instead of five or maybe seven? And why is the wavelength of 660 nm added as the third wavelength, not any other one in the red spectral region?
  12. Have the authors tried any machine learning algorithms? Why were only different vegetation indices computed using the three bands?
  13. How were the scattered points fitted in section Results should be provided in section 2.4 (including why this instead of that function was used in each case).
  14. Section 2.4.1, the references concerning NDVI, RVI, DVI etc. should be given here.
  15. Why is section 2.2 behind again after section 3??
  16. I’ve read the second section 2.3 for several times but still cannot figure out how was the saturation problem solved in this study. Could you introduce this with more details?

Author Response

First, we really appreciate your  helpful comments and valuable suggestions. According to the comments of the reviewers, we have made corrections and modifications as follows:

Reviewer #1

Point 1. Line 18: the existing two-band instrument: the single form is used here; do you indicate comparison with a single two-band instrument? If so, please provide its name and relevant wavelengths.

Response 1: This is a language problem. We referred to common two-band instruments here rather than a specific instrument. We have changed to plural form.

Point 2. Line 19: “relying on a single vegetation index”: this is incorrect. Even with only two wavelengths, different kinds of vegetation indices can be computed, eg. NDVI, DVI…

Response 2: This sentence has been revised. Our statement was incorrect here. What we meant was that two-band instruments could obtain fewer vegetation indices than three-band instruments.

Point 3. Line 19: “low accuracy of growth index retrieval”: this is inaccurate. I’m sure the performance is not always poor, please refine your statement.

Response 3: This sentence did not correctly express our meaning. We meant that the fitting accuracy was low when monitoring some of the growth indices. This sentence has been revised.

Point 4. Line 27: “could be used for monitoring the canopy vegetation index” index or indices, please make sure.

Response 4: The plural form is correct. The ‘index’ has been modified to the ‘indices’.

Point 5. Line 29: 0.64, 0.84 … 0.82, respectively.

Response 5: This sentence has been revised.

Point 6. Line 29, LAI, LDW, LNC, LNA: these abbreviations were not explained in their first appearance.

Response 6: These abbreviations have been explained in their first appearance.

Point 7. Line 35: do you mean that the RRMSE was improved by 0.22, 0.23 …? Note that the verb of the whole sentence was “improved”.

Response 7: Our statement was incorrect here. This sentence has been revised.

Point 8. Line 48, “they require destructive sampling are tedious, time-consuming”: with two verbs, this sentence doesn’t make sense.

Response 8: We have corrected this syntax problem.

Point 9. I’m surprised that the whole paragraph from line 52 to 65 said nothing about any airborne or satellite multispectral or hyperspectral sensors. If the authors want to confine their introduction to only portable sensors, state this clearly in the first place to avoid any misunderstanding.

Response 9: Thanks for the problem you pointed out, we realized that the original statement might be misleading. This article is based on a portable instrument, so we want to focus on introducing existing portable instruments in the section of Introduction. We have stated this at the beginning of this paragraph.

Point 10. Line 89-90, what do you mean by saying “only output the vegetation index”? Again, even for a two-band sensor, various kinds of indices can be computed. Also, as the authors suggested that the above-mentioned spectrometers were “costly, complicated to operate”, their approximate price, size, weight, as well as measurement manners etc. in comparison with the adopted system in this study should be clearly listed, emphasizing the importance of this new system (maybe in section 2.2). As far as I’m concerned, some of the mentioned systems (e.g. ASD) are not “complicated to operate”.

Response 10: Our statement was incorrect here. The sentence has been revised. More information of other systems has also been added in section 2.2. The complicated operation of ASD and other instruments mentioned in this article referred to general farmers. This sentence has been revised to avoid misunderstanding.

Point 11. Line 113-114, the authors said that two bands is not enough. Given so, why was only three bands adopted in this study? Why is three the optimal number, instead of five or maybe seven? And why is the wavelength of 660 nm added as the third wavelength, not any other one in the red spectral region?

Response 11. (1) The reason for why CGMD contained three bands referred to two factors. On the one hand, the information provided by the band reflectance should meet the needs of wheat growth monitoring. The previous generation product of the instrument, CGMD302, had poor monitoring accuracy for some crop growth indices, which drove us to increase the number of bands on this basis. On the other hand, when designing the instrument, we focused on its production cost. The number of bands has an important influence on the price of the instrument. Therefore, we tried to obtain as much information as possible with a small number of bands. Considering that vegetation indices constructed by predecessors were mostly two-band and three-band, and these three bands contain information of several growth indices, we set the number of bands to three.

(2) The reason why the 660 nm wavelength was added as the third wavelength came from the previous research results of our research group. As the absorption peak of crops, 660nm is more sensitive to growth indicators than other red light bands. Related references have been added.

Related references are as follows:

Zhu, Y.; Yao, X.; Tian, Y.; Liu, X.; Cao, W., Analysis of common canopy vegetation indices for indicating leaf nitrogen accumulations in wheat and rice. Int J Appl Earth Obs 2008, 10, (1), 1-10.

Point 12. Have the authors tried any machine learning algorithms? Why were only different vegetation indices computed using the three bands?

Response 12: The reasons why we only used the method of constructing vegetation indices were as follows.

(1) Vegetation indices are a simple, effective and empirical measure of crop growth status. Past researches have proved that there is a linear or non-linear relationship between vegetation indices and growth indices. Due to its ease of calculation, even farmers without relevant knowledge can master the method of constructing vegetation indices.

(2) When processing hyperspectral or other multi-feature data, we would use SPA-PLS, PCA, Regression Tree and other methods to process the data. Considering that only three bands’ information was obtained in this study, and the number of samples was less than 100. In the case of limited number of features and samples, other machine learning methods may not exhibit better results.

Point 13. How were the scattered points fitted in section Results should be provided in section 2.4 (including why this instead of that function was used in each case).

Response 13: More information has been added to section 2.4.

Point 14. Section 2.4.1, the references concerning NDVI, RVI, DVI etc. should be given here.

Response 14: The references concerning NDVI, RVI, DVI etc. have been given in section 2.4.1.

Point 15. Why is section 2.2 behind again after section 3??

Response 15: There is an error in the serial number after 3.1. This problem has been corrected

Point 16. I’ve read the second section 2.3 for several times but still cannot figure out how was the saturation problem solved in this study. Could you introduce this with more details?

Response 16: Our method was essentially to classify the data set to better distinguish the difference between growth indices under high biomass conditions. The difference in wheat canopy vegetation indices was small, even if the growth index was extremely different under conditions of moderate-to-high canopy biomass, which led to poor prediction accuracy of samples with high biomass. As shown in the figure 1(A), as the value of the abscissa increased, the distribution of scattered points became more discrete. This problem was difficult to solve with a single spectral monitoring model. Based on the differences of wheat growth indices in different phenological periods, the samples could be classified, which could reduce the gap between the values of scattered points with high biomass in each model and improve the prediction accuracy of monitoring models. As shown in the Figure 2(A), the performance of the models constructed based on the growth stages had been improved, especially in the case of high biomass. The verification result also showed that the fitted trend line of the predicted value and observed value was closer to the 11 line(y=0.8362x+0.253), as shown in Figure 2(B). This model construction method is simple and practical, and farmers can quickly learn to how to use it.

We have introduced our method with more details in section 3.3.

Images cannot be displayed in the box.Please see the attachment.

We hope that the revised manuscript could satisfy you and the requirements for publication in the journal. Thank you and the reviewers again for your help.

Yours sincerely,

Huaimin Li and Jun Ni

Nanjing Agricultural University

No.1 Weigang Road

Nanjing, Jiangsu 210095

P.R.China

Reviewer 2 Report

The work has been revised and, from my point of view, has improved and is suitable for publication in this journal. However, I would like to make some comments

Line 28. Delete repetition

Line 29. Indicate the meaning of the acronyms

Line 36. The RRMS value = 0.88 seems too high. Could you check it out?

Line 69. A parenthesis is missing

Line 198. Missing reference from Kjeldahl method?

It would be convenient to distinguish between the units used to express the different evaluated indices.

The number of measurements used for the linear regressions and to obtain the models should be indicated in the text or in the figures.

Author Response

First, we really appreciate you and the reviewers’ helpful comments and valuable suggestions. According to the comments of the reviewers, we have made corrections and modifications as follows:

Reviewer #2

Point 1. Line 28. Delete repetition

Response 1: We must admit that this sentence was misleading. The best vegetation index of LNC and LNA are both NDVI (R730, R815), so it looked like there was a ‘repetition’ here. This sentence has been modified to avoid misunderstanding.

Point 2. Line 29. Indicate the meaning of the acronyms.

Response 2: These abbreviations have been explained in their first appearance.

Point 3. Line 36. The RRMS value = 0.88 seems too high. Could you check it out?

Response 3: Thanks for your correction. The correct value is 0.28.

Point 4. Line 69. A parenthesis is missing

Response 4: The error here has been revised.

Point 5. Line 198. Missing reference from Kjeldahl method?

Response 5: The reference of Kjeldahl method has been added.

Point 6. It would be convenient to distinguish between the units used to express the different evaluated indices.

Response 6: Thanks for your suggestion. Units of different growth indices have been indicated in this manuscript. The evaluation indices RRMSE and R2 are dimensionless, so they have no units.

Point 7. The number of measurements used for the linear regressions and to obtain the models should be indicated in the text or in the figures.

Response 7: The number of samples has been indicated in the text. The reason why we did not indicate the number of samples in figures was that most figures contain multiple vegetation indexes. Too many numbers may cause misunderstandings and it is also unclear.

We hope that the revised manuscript could satisfy you and the requirements for publication in the journal. Thank you and the reviewers again for your help.

Yours sincerely,

Huaimin Li and Jun Ni

Nanjing Agricultural University

No.1 Weigang Road

Nanjing, Jiangsu 210095

P.R.China

Round 3

Reviewer 1 Report

The authors have answered my previous questions, and I don't have any more towards this manuscript.